materials science/nanotechnology

mesoporous, ZnO thin film, thermoelectric, Au nanoparticle, Al doping

**Author for correspondence:**
Hyung-Ho Park
e-mail: hhpark@yonsei.ac.kr

This article has been edited by the Royal Society of Chemistry, including the commissioning, peer review process and editorial aspects up to the point of acceptance.

# The thermoelectric properties of Au nanoparticle-incorporated Al-doped mesoporous ZnO thin films

Min-Hee Hong, Wooje Han, Kyu-Yeon Lee and Hyung-Ho Park

Department of Materials Science and Engineering, Yonsei University, Seoul 03722, Republic of Korea

(iD) H-HP, 0000-0001-5540-5433

Mesoporous Al-doped ZnO thin films incorporated with gold nanoparticles (Au NPs) were synthesized using a sol–gel and evaporation-induced self-assembly process. In this study, the complementary effects of Au NP incorporation and Al doping on the thermoelectric properties of mesoporous ZnO thin films were analysed. The incorporated Au NPs induced an increase in electrical conductivity but a detriment in the pore arrangement of the mesoporous ZnO thin film, which was accompanied by a decrease in porosity. However, the addition of the Al dopant minimized the pore structural collapse because of the inhibition of the grain growth in the ZnO skeletal structure, resulting in the enhancement of the pore arrangement and porosity. When the Au NPs and Al dopant were added at the same time, the degradation in the pore structure was minimized and the electrical conductivity was effectively increased, but the absolute value of the Seebeck coefficient was decreased. However, as a result, the thermoelectric power factor was increased by 2.4 times compared to that of the pristine mesoporous ZnO thin film. It was found that co-introducing the Au NPs and Al doping to the mesoporous ZnO structure was effective in preserving the pore structure and increasing the electric conductivity, thereby enhancing the thermoelectric property of the mesoporous ZnO thin film.

## 1. Introduction

Nowadays, fossil fuels continue to be rapidly depleted, and so the development of alternative energy sources has become very important and the interest in energy conversion techniques is rapidly increasing. Among the many energy conversion processes,

thermoelectricity has been actively studied, because this technique comprises a green energy conversion system in which electric current is generated from materials with a temperature gradient.

The thermoelectric property can be represented by the figure of merit,

$$ZT = \frac{S^2 \sigma T}{\kappa},$$

where $S$, $\sigma$, $\kappa$ and $T$ are the Seebeck coefficient, electrical conductivity, thermal conductivity and temperature, respectively. To enhance the thermoelectric property, a high Seebeck coefficient value, electrical conductivity and low thermal conductivity are essential.

However, there is a limit to increasing the ZT value because the Seebeck coefficient, electrical conductivity and thermal conductivity have an organic correlation. The Seebeck coefficient and electrical conductivity are in inverse relationship, and electrical conductivity and thermal conductivity are in a proportional relationship [1–3]. Therefore, in this study, the mesoporous structure was introduced to maximize the drop in the thermal conductivity, while minimizing the decrease in the electrical conductivity, and to increase the thermoelectric properties. The mesoporous structure is a structure in which pores in the range of 2–50 nm are distributed inside the material [4], and it was first reported by Mobil in 1992 [5]. The pore structure acts as a phonon scattering centre due to the smaller size of pores and shorter distance between pores than the phonon mean free path, and drastically decreases the thermal conductivity term by phonon ($\kappa_{ph}$). In addition, the open-pore structure has very large specific surface area and can be applied to various devices [6,7]. Their properties could be controlled by the pore structure such as pore size, porosity and pore distribution in the mesoporous structure [8]. Electrical conductivity is a term related to electron transport, which is influenced by carrier concentration and mobility. However, thermal conductivity is divided into by electron ($\kappa_{el}$) and phonon ($\kappa_{ph}$). When a mesoporous structure is introduced, $\kappa_{ph}$ could be selectively decreased by the introduction of the mesoporous structure, and the thermoelectric properties could be enhanced.

An ordered mesoporous structure has been synthesized using an evaporation-induced self-assembly (EISA) process in which a micelle structure was formed above the critical micelle concentration [9]. After annealing, the organic micelle structure was decomposed and the ordered mesoporous structure synthesized. However, the use of the mesoporous structure was limited in its applicability as a thermoelectric material due to its low electrical conductivity. Moreover, the independent control of electrical conductivity and thermal conductivity is essential for enhancing the thermoelectric properties. In this work, a doping process and the incorporation of metal nanoparticles (NPs) were adopted to increase the electrical conductivity of the mesoporous structure while maintaining the structural properties of the pores.

Zinc oxide (ZnO) is used in various applications such as gas sensors [10,11], and photovoltaic [12] and solar cells [13], because it is an n-type wide band gap semiconductor with good electrical, optical and photonic properties. In this work, sol–gel processing was adopted to use an EISA process. Sol–gel processing has many advantages such as low cost and ease of doping or nanoparticle incorporation. However, it is difficult to synthesize mesoporous ZnO thin films with an ordered structure because the crystallization temperature of ZnO with a hexagonal wurtzite structure is very low (300°C) [14], and the crystalline growth of the ZnO skeletal structure inhibits the regular formation of ordered pores [15].

Moreover, the high reactivity of the Zn ion precursor is also a hurdle that must be overcome when synthesizing an ordered mesoporous structure [4,15], and the low electrical conductivity of mesoporous structure also demerits its value when applied to thermoelectric devices. Hence, in this work, Au NPs and Al as a dopant were incorporated into the mesoporous ZnO structure. The incorporation of the Au NPs increased the electrical conductivity of the mesoporous ZnO thin film due to surface plasmon effects [16] and increased the electrical conductivity with a minimal increase in the thermal conductivity.

However, the incorporation of Au NPs had a deleterious effect because of pore structural collapse [17,18], so Al was added as a dopant because the Al ions were substituted for Zn ions at the lattice sites. This induced stress in the thin film, which distorted the crystalline structure and as a result, grain growth was inhibited and the pore structure was maintained [19]. The grain growth of ZnO hexagonal wurtzite structure was inhibited because of the difference in the radius of $Al^{3+}$ (0.054 nm) and $Zn^{2+}$ (0.074 nm) [20]. That is to say, the pore structural collapse due to the incorporation of Au NPs was depressed by doping with Al, while still maintaining the increase in electrical conductivity. When the ZnO structure was doped with Al, the electrical conductivity of Al-doped ZnO (AZO) was increased because of an increase in carrier concentration [14]. In addition, the complementary effect of the Al doping and incorporating the Au NPs on the crystallization, pore structure, porosity and thermoelectric property of the mesoporous ZnO thin films was investigated.

# 2. Experimental procedure

SiO$_2$/Si substrates were used for the preparation of the pristine mesoporous ZnO, mesoporous ZnO incorporated with Au NPs (hereafter, Au NPs-ZnO) and mesoporous AZO incorporated with Au NPs (hereafter, Au NPs-AZO) thin films. In this paper, SiO$_2$/Si substrate was prepared by setting the size to $2 \times 2$ cm. The substrates were cleaned using acetone, ethanol and de-ionized water sequentially; the substrates were dipped and cleaned by ultrasound for 15 min in each solvent. In this work, zinc acetate dihydrate [Zn(CH$_3$COOH)$_2$·H$_2$O], n-propanol, Brij-S10 (C$_{58}$H$_{118}$O$_{21}$, Aldrich, MW 711), aluminium nitrate nonahydrate [Al(NO$_3$)·9H$_2$O], chloroauric acid (HAuCl$_4$) and monoethanolamine (MEA) were used as Zn precursor, solvent, surfactant, dopant source, Au precursor and complex agent, respectively.

In this work, the MEA/Zn molar ratio was fixed at 1 and the surfactant/Zn ratio at 0.05. Similarly, the molar ratios of the Al and Au precursors to Zn precursor were fixed at 0.02 (i.e. the atomic ratio of both Al to Zn and Au to Zn was 2 at%). The molar ratio of zinc acetate dihydrate : Brij-S10 : MEA : n-propanol was $1 : 0.05 : 1 : 34.5$. After 1 day of stirring to achieve stabilization, the solutions were spin-coated onto the SiO$_2$/Si substrate at 3000 r.p.m. for 30 s. To remove any residual organics, the as-prepared thin films were preheated at 300°C for 10 min. The pristine mesoporous ZnO, Au NPs-ZnO and Au NPs-AZO thin films were then annealed at 450°C for 4 h.

The crystallinity of the thin films was investigated using X-ray diffraction (XRD) with Cu K$\alpha$ radiation ($\lambda = 1.5418$ Å) at angles of 20°–60°. At the Pohang Light Source (PLS) in Korea, a grazing incidence small-angle X-ray scattering (GISAXS) analysis was progressed at the 3C beamline ($\lambda = 1.54$ Å and $\Delta\lambda/\lambda = 5 \times 10^{-4}$) to analyse the pore structure [21]. The shape and size of the Au NPs were analysed using transmission electron microscopy (TEM, JOEL JEM-2100F) at 300 kV and energy dispersive spectroscopy (EDS). TEM sampling was achieved by carbon-coated Cu grid by dispersing the powder in ethanol instead of ion milling or scratching. The powder was formed by heat treatment of sol solution. The porosity of the films was analysed using an ellipsometer (Gatan L117C, 632.8 nm He–Ne laser) and calculated using the Lorentz–Lorenz equation [22]. Finally, the thermoelectric properties of the Au NPs-ZnO and Au NPs-AZO thin films were measured by detecting the Seebeck voltage with a SEEPEL thermoelectric properties measurement system (TEP 850). The instrument allows measurement of film samples. The Seebeck coefficient is defined as the ratio of the voltage difference to the temperature difference between two sides of the materials ($S = \Delta V / \Delta T$). The temperature range could be set by using a software system [19]. In this paper, the temperature difference was from 323 to 478 K at intervals of 50 K under a helium gas flow atmosphere.

# 3. Results and discussion

In this work, HAuCl$_4$ was used as the precursor of the Au NPs; TEM/EDS analyses were carried out to confirm the formation of the latter in the mesoporous ZnO thin films. Figure 1 shows the TEM and EDS results of the pristine ZnO and Au NPs-AZO samples. The pores were not clearly distinguishable because of planar overlapping in the two-dimensional TEM image but in both images, several tens of nanometre-sized pores were well observed, especially with pristine ZnO sample (figure 1$a$). In the TEM image of Au NPs-AZO sample (figure 1$b$), the presence of the Au NPs at a size of almost 20 nm was clearly confirmed. The comparison data of EDS results of both samples also showed the existence of the Au NPs in the mesoporous Au NPs-AZO sample.

The crystalline structures of the pristine mesoporous ZnO, Au NPs-ZnO and Au NPs-AZO thin films were analysed using XRD. As shown in figure 2, regardless of Al doping and whether the Au NPs had been incorporated or not, all of the mesoporous ZnO thin films had a hexagonal wurtzite structure. The XRD 2$\theta$ peaks at 31.8°, 33.4° and 36.2° were indexed as the 100, 002 and 101 planes, respectively. In addition, the Au (111) peak was confirmed at 38.2° in the cases of incorporated Au NPs in the mesoporous ZnO thin films.

Using the Scherrer equation [23], the size of the Au NPs was calculated with the full-width at half-maximum value of the Au (111) peak. The Scherrer equation is shown below.

$$D = \frac{K\lambda}{\beta\cos\theta}.$$

Here, $\beta$ is the width of the observed diffraction line at its half intensity maximum, $K$ is the shape factor set to 0.9 and $\lambda$ is the wavelength of Cu K$\alpha$. The size of Au NPs in AZO and ZnO was 20.7 nm

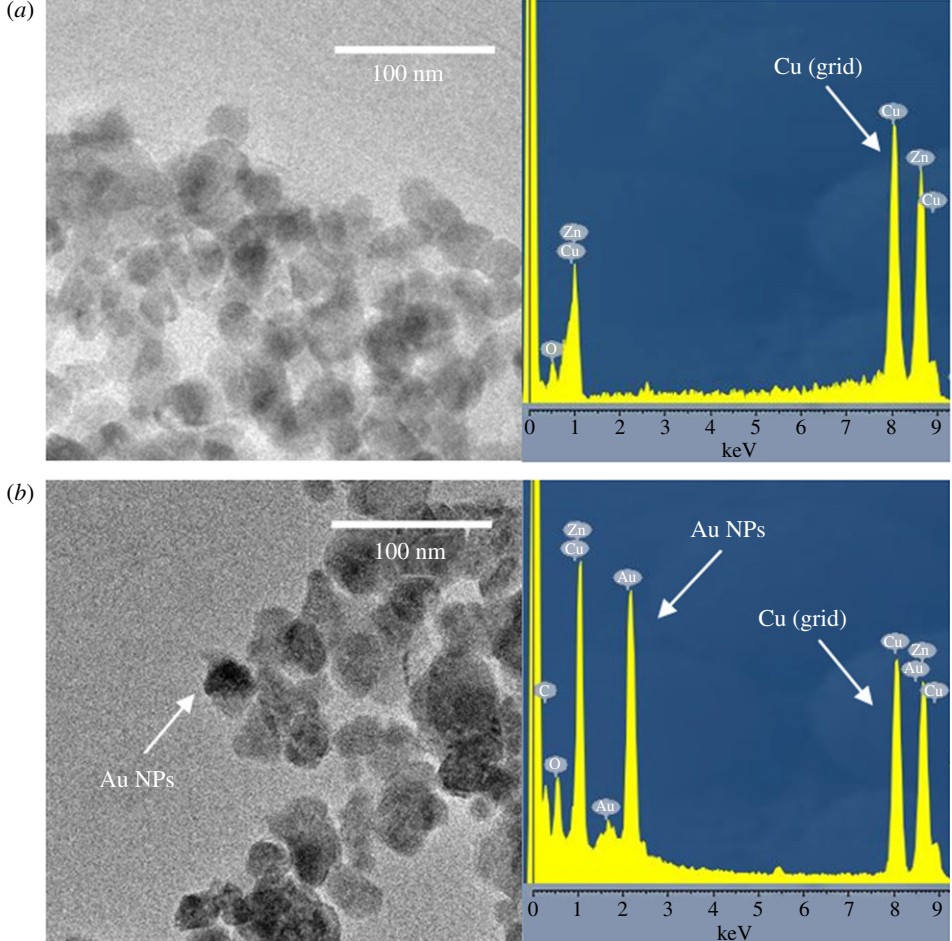

**Figure 1.** TEM images and EDS results of (*a*) pristine ZnO and (*b*) Au NPs-AZO samples.

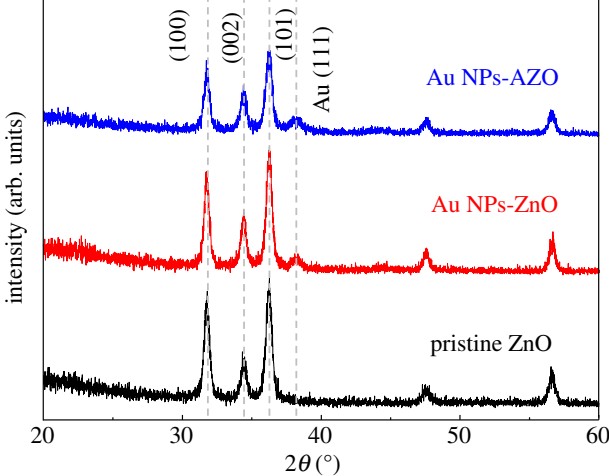

**Figure 2.** XRD patterns of the pristine mesoporous ZnO, Au NPs-ZnO and Au NPs-AZO thin films.

and 21.2 nm, respectively which matched well with the TEM observations in figure 1. Furthermore, the intensity of the XRD peaks of the ZnO thin film incorporated with Au NPs was almost the same when compared with that of the pristine ZnO thin film. This result indicates that the incorporation of the Au NPs did not affect the crystallization and grain growth in the ZnO skeleton with a wurtzite structure in the mesoporous thin film.

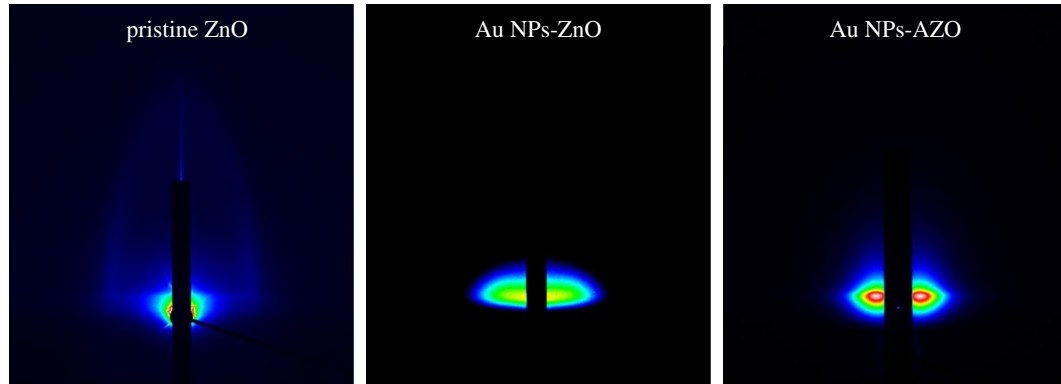

**Figure 3.** GISAXS patterns of the pristine mesoporous ZnO, Au NPs-ZnO and Au NPs-AZO thin films.

However, as shown in figure 2, the grain growth in the ZnO skeletal structure was inhibited by the Al doping, as indicated by a reduction in the XRD peak intensity of the Au NPs-AZO thin film compared with that of the pristine mesoporous ZnO and Au NPs-ZnO thin films. The decrease in XRD peak intensity was accompanied by a radius difference between $Al^{3+}$ (0.054 nm) and $Zn^{2+}$ (0.074 nm). The formation of hexagonal wurtzite structure for ZnO caused by sol–gel processing and an annealing temperature of 400°C has been reported [24], which is lower than in our experiments. After crystallization, the pore structure could have collapsed as the grain growth progressed, but by doping the ZnO structure with Al, the grain growth was effectively inhibited compared with that of the pristine ZnO thin film [18]. The Al doping affected the mesoporous ZnO structure by preventing the collapse of the pore structure due to the Au NP incorporation and the grain growth in the ZnO skeletal structure, as was evident in the Au NPs-ZnO thin film without the dopant.

In a mesoporous structure, the thermal conductivity is decreased because the pores act as phonon scattering sites. Thus, maintaining the porosity and pore structural ordering of the pristine mesoporous ZnO structure during the Au NP incorporation and Al doping was important to enhance the thermoelectric properties of the films. In this work, to investigate the pore structural change of the mesoporous ZnO composite thin films caused by the Au NP incorporation and the Al doping, a GISAXS analysis was carried out, the results of which are given in figure 3. In a GISAXS analysis, when the X-rays are scattered by the sample surface at a low grazing incidence, X-rays are emitted due to a change in the surface density. The emitted X-rays strike the two-dimensional detector (horizontal and vertical) and a GISAXS wing pattern is obtained when an ordered structure is present in a thin film [25].

As shown in figure 3, a GISAXS wing pattern was obtained with the pristine mesoporous ZnO thin film from the regularly distributed, ordered pores. However, the wing pattern changed with the Au NP incorporation and the Al doping. When the Au NPs were incorporated, the GISAXS wing pattern disappeared because the Au NPs invade pore structure, causing it to collapse. When the pore structure was lost due to disordering or collapse of the pores, the GISAXS wing pattern disappeared. However, the wing pattern was faintly observable for the Au NPs-AZO thin film, indicating a partial ordering of the pores. From these results, it could be said that the Au NP incorporation had a hindering effect on the ordering of the pores, but the Al doping induced an enhancement in pore ordering in the mesoporous ZnO thin film. This enhancement due to the Al doping might have been caused by the grain growth inhibition due to Al substituting for Zn in the ZnO structure. As shown in the XRD results, the Al doping resulted in a reduced XRD peak intensity due to the inhibited grain growth of ZnO skeletal structure, and this inhibition preserved the pore structure of mesoporous ZnO, i.e. the ordering of the pores. This preserved ordering of the pores resulted in the GISAXS wing pattern.

To check for pore collapse caused by the Au NP incorporation and the ordering of the pores due to Al doping, the porosities of the pristine mesoporous ZnO, Au NPs-ZnO and Au NPs-AZO thin films were measured and the results calculated using the Lorentz–Lorenz equation [22] (figure 4),

$$1 - F_p = \frac{(n_f^2 - 1)/(n_f^2 + 2)}{(n_s^2 - 1)/(n_s^2 + 2)} \, ,$$

where $F_p$ is the pore volume fraction, $n_f$ is the refractive index of the film and $n_s$ is the refractive index of air. Porosity could be calculated from the difference between the refractive index of the original material

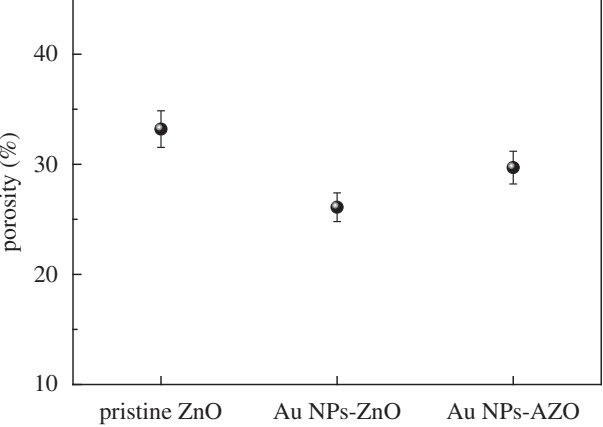

**Figure 4.** The porosity values of the pristine mesoporous ZnO, Au NPs-ZnO and Au NPs-AZO thin films.

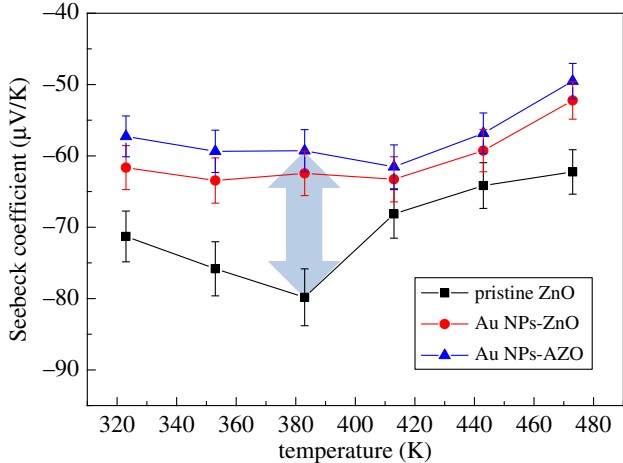

**Figure 5.** The Seebeck coefficient values of the pristine mesoporous ZnO, Au NPs-ZnO and Au NPs-AZO thin films according to temperature.

and the refractive index measured after the structure was synthesized. In this paper, the refractive index of ZnO is 2.1 [26], and for ZnO with Au NPs was 2.081. The ratio of ZnO to Au was calculated and estimated [18]. The measured refractive index was 1.629 for pristine ZnO, 1.695 for Au NPs ZnO and 1.652 for Au NPs-AZO. Porosity was calculated based on the refractive index value. The porosity of the mesoporous ZnO decreased with Au NP incorporation from 33.2% to 26.1% due to the invasion of the pores by the Au NPs, causing the pore structure to collapse. This decreasing behaviour is in good agreement with the disappearance of the GISAXS wing pattern for the Au NPs-ZnO film, as shown in figure 3. However, in the case of the Al doping of the Au NPs-ZnO thin film, the porosity increased from 26.1 to 29.7%, corresponding to resistance to the pore structure collapse due to the Au NPs, i.e. a tendency to maintain the ordered pore structure found in the pristine ZnO film. This increment was followed by grain growth inhibition of ZnO structure due to Al doping as confirmed in XRD observation result (figure 2) and this result was well matched by the GISAXS observation of the Au NPs-AZO thin film shown in figure 3. To increase the thermoelectric property, independent control of both thermal conductivity and electrical conductivity is essential. Thus, in this work, the maintenance of porosity and pore arrangement was very important with Au NP and/or dopant incorporation to enhance the electrical conductivity. From the XRD and GISAXS results, it can be said that the pore structural distortion due to the Au NP incorporation in mesoporous ZnO was minimized by Al doping. As a result, through the Al doping and the Au NP incorporation, the increase in thermal conductivity was minimized while the increase in electrical conductivity was maximized.

Figure 5 shows the Seebeck coefficient change in the pristine mesoporous ZnO, Au NPs-ZnO and Au NPs-AZO thin films. Because ZnO is an n-type semiconductor, the films attained negative Seebeck coefficient values. As shown in figure 5, the pristine mesoporous ZnO thin film had the largest

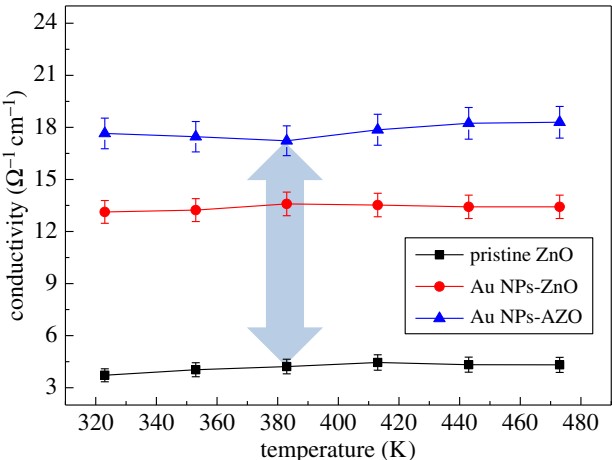

**Figure 6.** The electrical conductivity of the pristine mesoporous ZnO, Au NPs-ZnO and Au NPs-AZO thin films according to temperature.

absolute Seebeck coefficient value along with the lowest electrical conductivity, and thus the smallest carrier concentration. The Seebeck coefficient is expressed as

$$S = \left(\frac{8\pi^2 k_B^2}{3eh^2}\right) m^* T \left(\frac{\pi}{3n}\right)^{2/3},$$

where $m^*$ is the effective mass, $k_B$ is Boltzmann's constant, $h$ is Planck's constant, $T$ is the absolute temperature and $n$ is the carrier concentration [27]. Hence, when the Au NPs were incorporated into the mesoporous ZnO thin film, the absolute Seebeck coefficient value decreased compared to the pristine ZnO thin film. Moreover, after Al doping was simultaneously applied to the Au NPs-ZnO thin film, the resulting film attained the smallest Seebeck coefficient absolute value accompanied by an increase in electrical conductivity. Because the Au NP incorporation caused the surface plasmon effect and the Al doping induced an increase in carrier concentration, the Seebeck coefficient value decreased in both the Au NPs-ZnO and Au NPs-AZO thin films [16]. For the pristine mesoporous ZnO thin film, the highest Au Seebeck coefficient value was $79.8\ \mu V\ K^{-1}$ at 383 K, whereas $59.3\ \mu V\ K^{-1}$ was obtained for Au NPs-AZO thin film. That is to say, under these experimental conditions, a maximum reduction in the Seebeck coefficient value was around 25% in the mesoporous ZnO thin film due to the Au NP incorporation and Al doping. Especially for the reduction due to the Al doping, the difference in the Seebeck coefficient values between the Au NPs-ZnO and Au NPs-AZO films was less than 10% (from 62.4 to $59.3\ \mu V\ K^{-1}$).

The electrical conductivity of the pristine mesoporous ZnO, Au NPs-ZnO and Au NPs-AZO thin films is given in figure 6. The pristine mesoporous ZnO thin film showed relatively low electrical conductivity because each pore acts as an electron scattering centre. At 383 K, the same as for the Seebeck coefficient measurements, the electrical conductivity increased by around 220% (from 4.2 to $13.4\ \Omega^{-1}\ cm^{-1}$) due to the surface plasmon effect of the Au NPs [16]. Moreover, a further increase of around 33% (from 13.4 to $17.8\ \Omega^{-1}\ cm^{-1}$) was also observed after the Al doping due to an increase in carrier concentration.

As shown in figures 2 and 3, Al doping induced degradation in crystallinity and a decrease in grain size. Usually, worse electrical conductivity is obtained with crystallinity degradation and a grain size decrease. However, the electrical conductivity of the Au NPs-AZO thin film rather increased with a small decrease in the Seebeck coefficient. Of most importance is that the pore structure of the mesoporous ZnO thin film and its intrinsic nature of limiting heat transfer was effectively maintained through the Al doping [18]. It can be said that the thermoelectric properties of Au NP-incorporated mesoporous ZnO thin films were enhanced by the Al doping by minimizing the decrease in the Seebeck coefficient value and improving the pore arrangement while maximizing the increase in electrical conductivity. When considering the change in the figure of merit, ZT, as a thermoelectric property, an increase by almost 2–2.4 times was calculated in the power factor ($S^2\sigma$) for the Au NPs-ZnO and Au NPs-AZO thin films compared to that of the pristine mesoporous ZnO thin film without considering the change in thermal conductivity, $\kappa$. However, the porosity and pore structure were maintained with Al doping of the Au NPs-ZnO thin film, and this should be given more emphasis than the enhancement in ZT. Hence, the Au NP-incorporated

Al-doped mesoporous ZnO could be a good candidate thermoelectric material due to high electrical conductivity and enhanced pore arrangement.

# 4. Conclusion

In this work, an Au NP-incorporated, Al-doped mesoporous ZnO thin film was synthesized using a sol–gel and EISA process, and the change in the thermoelectric property of the resulting thin film based on crystallinity, pore structure, the Seebeck coefficient and electrical conductivity was investigated. The Au NP incorporation was found to induce not only an increase in the electrical conductivity but also a decrease in the Seebeck coefficient value and degradation of the pore structure of the resulting thin film. However, Al doping contributed toward maintaining the pore structure due to the inhibition of grain growth and limiting the reduction in porosity. It can be said that Al doping and Au NP incorporation both contributed toward increasing the electrical conductivity of the mesoporous ZnO by compensating for each other's effects on the crystallinity and pore structure of the ZnO. Besides, Al doping of the Au NPs-ZnO thin film at 2 at% recovered 50% of the porosity decrease caused by the Au NPs (from 33.2% via 26.1% to 29.7%), and resulted in a gradual increase in power factor from almost 2 to 2.4 times compared to that of the pristine mesoporous ZnO thin film. Al doping and Au NP incorporation enhanced the thermoelectric property of the mesoporous ZnO while minimizing the pore structure degradation and maximizing the electrical conductivity.

Data accessibility. Data available from the Dryad Digital Repository: https://doi.org/10.5061/dryad.g2j4t50 [28].

Authors' contributions. M.-H.H. designed the experiments, and performed the sample fabrication, characterization and measurements with the analysis. W.H. and K.-Y.L. performed the sample characterization and measurements. H.-H.P. conceived the idea of the research and supervised the experiments, analyses and manuscript. All the authors discussed the progress of the research and reviewed the manuscript.

Competing interests. We declare have no competing interests.

Funding. This work was supported by Nano-Convergence Foundation (www.nanotech2020.org) funded by the Ministry of Science, ICT and Future Planning (MSIP), Korea and the Ministry of Trade, Industry and Energy (MOTIE), Korea (Project Number: R201602310). The experiments at PLS were supported, in part, by MEST and POSTECH. This work was supported (researched) by the third Stage of Brain Korea 21 Plus Project in 2019.

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
