## [Reviewer comments · Royal Society Open Science]

Review History

RSOS-181799.R0 (Original submission)

Review form: Reviewer 1

Is the manuscript scientifically sound in its present form?

Yes

Are the interpretations and conclusions justified by the results?

Yes

Is the language acceptable?

Yes

Is it clear how to access all supporting data?

Yes

Do you have any ethical concerns with this paper?

No

Have you any concerns about statistical analyses in this paper?

No

Recommendation?

Major revision is needed (please make suggestions in comments)

Comments to the Author(s)

Comments on RSOS-181799

The present study reported a superior thermoelectric property of ZnO thin films by the assistance of Au nanoparticles and Al-doping. The topic is of general interest to the community, and the conclusions were supported by the experimental results. However, the manuscript still suffers from some problems before its publication. Detailed comments are listed below for further improvement:

1. The introduction section should be significantly enhanced. I can see the necessity of the study, but the novelty of the study was not well illustrated because a brief introduction of the state of art peer researches, especially mesoporous thermoelectric materials, was missing.
2. Though I can manage to follow the message in the introduction section, the logic in some places is rather confusing, e.g., line 46, page 1 "Hence, to control these factors...". Please check the entire section.
3. Please provide the fundamental data such as the porosity, or size of the SiO₂/Si substrates.
4. I believe there was a mistake about the sentence of "mesoporous ZAO incorporated with Au NPs (hereafter, Au NPs-AZO) thin films)".
5. Detailed measurement process about the thermoelectric properties should be provided.
6. TEM images for both the pristine mesoporous ZnO and mesoporous ZnO incorporated with Au NPs should be provided for comparison.
7. I believe EDX mapping results of the samples may also provide more valuable information about the modification quality.
8. Please provide the Scherrer equation and size calculation process. Also only one Au NPs was indicated in Fig. 1, which cannot provide a solid support for the Scherrer calculation results.
9. Error bars should be provided in Fig. 4~6 to indicate the statistical significance, otherwise it is hard to determine the observed differences.

Review form: Reviewer 2 (Mahendra More)**Is the manuscript scientifically sound in its present form?**

No

Are the interpretations and conclusions justified by the results?

Yes

Is the language acceptable?

Yes

Is it clear how to access all supporting data?

Yes

Do you have any ethical concerns with this paper?

No

Have you any concerns about statistical analyses in this paper?

I do not feel qualified to assess the statistics

Recommendation?

Accept with minor revision (please list in comments)

Comments to the Author(s)

1. Figure 1 depicts the TEM image and EDS results of the Au NPs-AZO thin film, which reveals formation of Au nanoparticles only. How sample was prepared for TEM analysis? By ion milling or scratching-off the material from Si/SiO₂ substrate, dispersing it in some solvent and drop casting it on Cu grid? TEM images of pristine mesoporous ZnO and Au-NPs-ZnO should be provided for better understanding. In addition, the EDAX spectrum shows presence of Zn, Au, O, and Cu atomic species. There is no peak due to Al. Why? It is better to provide the atomic percentage of these elements in tabular form.
2. The authors have recently published similar work in Materials Chemistry and Physics 212 (2018) 499 (Ref 17). What is the novelty in the present work? Whether the earlier results (Ref 17) are superior (or inferior) to the present ones?
3. What is effect of size of Au NPs on ZT? As the surface plasmon resonance depends on size and shape of the NPs, will it influence the ZT?
4. The manuscript contains a lot many typos and grammatical mistakes. The manuscript should be re-written carefully, avoiding vague statements. (For example, the statement "...the pore structural distortion due to the Au NP incorporation in mesoporous ZnO to enhance electrical conductivity was minimized by Al doping" seems unclear.)

Decision letter (RSOS-181799.R0)

25-Jan-2019

Dear Professor Park:

Title: The thermoelectric properties of Au nanoparticle-incorporated Al-doped mesoporous ZnO thin films

Manuscript ID: RSOS-181799

The editor assigned to your manuscript has now received comments from reviewers. We would like you to revise your paper in accordance with the referee and Subject Editor suggestions which can be found below (not including confidential reports to the Editor). Please note this decision does not guarantee eventual acceptance.

Please submit your revised paper before 17-Feb-2019. Please note that the revision deadline will expire at 00.00am on this date. If we do not hear from you within this time then it will be assumed that the paper has been withdrawn. In exceptional circumstances, extensions may be possible if agreed with the Editorial Office in advance. We do not allow multiple rounds of revision so we urge you to make every effort to fully address all of the comments at this stage. If deemed necessary by the Editors, your manuscript will be sent back to one or more of the original reviewers for assessment. If the original reviewers are not available we may invite new reviewers.

• Funding statement

Please include a funding section after your main text which lists the source of funding for each author.

RSC Associate Editor:
Comments to the Author:
(There are no comments.)

RSC Subject Editor:
Comments to the Author:
(There are no comments.)

Reviewers' Comments to Author:
Reviewer: 1

Comments to the Author(s)
Comments on RSOS-181799

The present study reported a superior thermoelectric property of ZnO thin films by the assistance of Au nanoparticles and Al-doping. The topic is of general interest to the community, and the

conclusions were supported by the experimental results. However, the manuscript still suffers from some problems before its publication. Detailed comments are listed below for further improvement:

1. The introduction section should be significantly enhanced. I can see the necessity of the study, but the novelty of the study was not well illustrated because a brief introduction of the state of art peer researches, especially mesoporous thermoelectric materials, was missing.
2. Though I can manage to follow the message in the introduction section, the logic in some places is rather confusing, e.g., line 46, page 1 "Hence, to control these factors...". Please check the entire section.
3. Please provide the fundamental data such as the porosity, or size of the SiO₂/Si substrates.
4. I believe there was a mistake about the sentence of "mesoporous ZAO incorporated with Au NPs (hereafter, Au NPs-AZO) thin films)".
5. Detailed measurement process about the thermoelectric properties should be provided.
6. TEM images for both the pristine mesoporous ZnO and mesoporous ZnO incorporated with Au NPs should be provided for comparison.
7. I believe EDX mapping results of the samples may also provide more valuable information about the modification quality.
8. Please provide the Scherrer equation and size calculation process. Also only one Au NPs was indicated in Fig. 1, which cannot provide a solid support for the Scherrer calculation results.
9. Error bars should be provided in Fig. 4~6 to indicate the statistical significance, otherwise it is hard to determine the observed differences.

Reviewer: 2

Comments to the Author(s)

1. Figure 1 depicts the TEM image and EDS results of the Au NPs-AZO thin film, which reveals formation of Au nanoparticles only. How sample was prepared for TEM analysis? By ion milling or scratching-off the material from Si/SiO₂ substrate, dispersing it in some solvent and drop casting it on Cu grid? TEM images of pristine mesoporous ZnO and Au-NPs-ZnO should be provided for better understanding. In addition, the EDAX spectrum shows presence of Zn, Au, O, and Cu atomic species. There is no peak due to Al. Why? It is better to provide the atomic percentage of these elements in tabular form.
2. The authors have recently published similar work in *Materials Chemistry and Physics* 212 (2018) 499 (Ref 17). What is the novelty in the present work? Whether the earlier results (Ref 17) are superior (or inferior) to the present ones?
3. What is effect of size of Au NPs on ZT? As the surface plasmon resonance depends on size and shape of the NPs, will it influence the ZT?
4. The manuscript contains a lot many typos and grammatical mistakes. The manuscript should be re-written carefully, avoiding vague statements. (For example, the statement "...the pore structural distortion due to the Au NP incorporation in mesoporous ZnO to enhance electrical conductivity was minimized by Al doping" seems unclear.)

Author's Response to Decision Letter for (RSOS-181799.R0)

See Appendix A.

RSOS-181799.R1 (Revision)

Review form: Reviewer 1

Is the manuscript scientifically sound in its present form?

Yes

Are the interpretations and conclusions justified by the results?

Yes

Is the language acceptable?

Yes

Is it clear how to access all supporting data?

Not Applicable

Do you have any ethical concerns with this paper?

No

Have you any concerns about statistical analyses in this paper?

No

Recommendation?

Accept as is

Comments to the Author(s)

Thank the authors for taking efforts to address my concerns. Now I believe the manuscript is publishable.

Decision letter (RSOS-181799.R1)

26-Mar-2019

Dear Professor Park:

Title: The thermoelectric properties of Au nanoparticle-incorporated Al-doped mesoporous ZnO thin films

Manuscript ID: RSOS-181799.R1

It is a pleasure to accept your manuscript in its current form for publication in Royal Society Open Science. I apologise that this took longer than usual. The chemistry content of Royal Society Open Science is published in collaboration with the Royal Society of Chemistry.

RSC Associate Editor:
Comments to the Author:
(There are no comments.)

RSC Subject Editor:
Comments to the Author:
(There are no comments.)

Reviewer(s)' Comments to Author:
Reviewer: 1

Comments to the Author(s)
Thank the authors for taking efforts to address my concerns. Now I believe the manuscript is publishable.

Appendix A

Reviewers' Comments to Author:

Reviewer: 1

Comments to the Author(s)

Comments on RSOS-181799

The present study reported a superior thermoelectric property of ZnO thin films by the assistance of Au nanoparticles and Al-doping. The topic is of general interest to the community, and the conclusions were supported by the experimental results. However, the manuscript still suffers from some problems before its publication. Detailed comments are listed below for further improvement:

Comment #1. The introduction section should be significantly enhanced. I can see the necessity of the study, but the novelty of the study was not well illustrated because a brief introduction of the state of art peer researches, especially mesoporous thermoelectric materials, was missing.

► The authors revised the Introduction part according to the comment of the reviewer.

Before revision (Results and discussion):

However, these factors have an interdependent relationship, for example in that the Seebeck coefficient value decreases and thermal conductivity increases as the electrical conductivity increases [1-3]. Hence, to control these factors independently, a mesoporous structure was adopted as a thermoelectric material in this work because it has low thermal conductivity.

A mesoporous structure has pores within the range 2 to 50 nm [4], and since one was first reported in 1992 [5], the analysis of these structures has markedly increased because of their low density, low thermal conductivity, high specific surface area, etc. [6,7]. Moreover, these properties can be changed depending on the pore size and pore arrangement of the mesoporous structure [8].

After revision (Results and discussion):

However, there is a limit to increasing the ZT value because the Seebeck coefficient, electrical conductivity, and thermal conductivity have an organic correlation. Seebeck coefficient and electrical conductivity in inversely relationship, and electrical conductivity and thermal conductivity are in a proportional relationship [1-3]. Therefore, in this study, the mesoporous structure was introduced to maximize the drop in the thermal conductivity while minimizing the decrease in the electrical conductivity, and to increase the thermoelectric properties. The mesoporous structure is a structure in which pores in the range of 2-50 nm are distributed inside the material [4], and it was first reported by Mobil in 1992 [5]. The pores structure acts as a phonon scattering center due to smaller size of pores and shorter distance between pores than phonon mean free path, and drastically decrease the thermal conductivity term by phonon (κ_{ph}). In addition, the open-pore structure has very large specific surface area and can be applied to various devices [6, 7]. Their properties could be controlled by pore structure such as pore size, porosity, and pore distribution in mesoporous structure [8]. Electrical conductivity is a term related to electron transport, which is influenced by carrier concentration and mobility. However, thermal conductivity is divided into term by electron (κ_{el}) and phonon (κ_{ph}). When a mesoporous structure is introduced, κ_{ph} could be selectively decreased by an introduction of mesoporous structure, and the thermoelectric properties could be enhanced.

Comment #2. Though I can manage to follow the message in the introduction section, the logic in some places is rather confusing, e.g., line 46, page 1 “Hence, to control these factors...”. Please check the entire section.

► As mentioned in comment # 1, authors modified the Introduction part.

Comment #3. Please provide the fundamental data such as the porosity, or size of the SiO₂/Si substrates.

► According to the comments of the reviewer, authors referred to the size of the SiO₂ / Si substrate in the “Experimental procedure” part and added the porosity estimation basis in the “Results and Discussion” part. The porosity values had been already given in the previously submitted manuscript.

After revision (Experimental Procedure):

In this paper, SiO₂ / Si substrate was prepared by setting the size to 2 cm x 2 cm.

After revision (Results and discussion):

Porosity could be calculated from the difference between the refractive index of the original material and the refractive index measured after the structure was synthesized. In this paper, the refractive index of ZnO is estimated to be 2.1[26], and for ZnO with Au NPs was estimated to be 2.081. The ratio of ZnO to Au was calculated and estimated. [18]. The measured refractive indices were 1.629 for pristine ZnO, 1.695 for Au NPs ZnO, and 1.652 for Au NPs-AZO. Porosity was calculated based on refractive index value. The porosity of the mesoporous ZnO decreased with Au NP incorporation from 33.2% to 26.1% due to the invasion of the pores by the Au NPs, causing the pore structure to collapse.

[26] F.K. Shan, Y.S. Yu, Band gap energy of pure and Al-doped ZnO thin films, J. Eur. Ceram. Soc. 24 (2004) 1869-1872.

Comment #4. I believe there was a mistake about the sentence of “mesoporous ZAO incorporated with Au NPs (hereafter, Au NPs-AZO) thin films”.

- ▶ According to the comment of the reviewer, typo was revised.

After revision (Experimental Procedure):

mesoporous AZO incorporated with Au NPs (hereafter, Au NPs-AZO).

Comment #5. Detailed measurement process about the thermoelectric properties should be provided.

- ▶ According to the comments of the reviewer, authors added detailed measurement process about thermoelectric properties in “Experimental procedure” part.

After revision (Experimental Procedure):

The instrument allows measurement of film samples. The Seebeck coefficient is defined as the ratio of the voltage difference to the temperature difference between two sides of the materials ($S=\Delta V/\Delta T$). The temperature range could be set by using software system [19]. In this paper, the temperature difference from 323 to 478 K at intervals of 50 K under a helium gas flow atmosphere.

Comment #6. TEM images for both the pristine mesoporous ZnO and mesoporous ZnO incorporated with Au NPs should be provided for comparison.

- ▶ According to the comments of the reviewer, authors added TEM image and EDS data.

After revision (Fig. 1):

Fig. 1. TEM images and EDS results of (a) pristine ZnO and (b) Au NPs-AZO samples.

Before revision (Results and discussion):

Fig. 1 shows the TEM and EDS results of the Au NPs-AZO thin film. In the TEM image, although the presence of the Au NPs at a size of almost 20 nm was confirmed, the pores were not clearly distinguishable because of planar overlapping in the two dimensional TEM image. The TEM and EDS results also clearly confirmed the existence of the Au NPs in the mesoporous AZO thin film.

Fig. 1. A TEM image and EDS results of the Au NPs-AZO thin film.

After revision (Results and discussion):

Fig. 1 shows the TEM and EDS results of the pristine ZnO and Au NPs-AZO samples. The pores were not clearly distinguishable because of planar overlapping in the two dimensional TEM image but in both images, several tens nanometer sized pores were well observed, especially with pristine ZnO sample (Fig. 1(a)). In the TEM image of Au NPs-AZO sample (Fig. 1(b)), the presence of the Au NPs at a size of almost 20 nm was clearly confirmed. The comparison data of EDS results of both samples also showed the existence of the Au NPs in the

mesoporous Au NPs-AZO sample.

Fig. 1. TEM images and EDS results of (a) pristine ZnO and (b) Au NPs-AZO samples.

Comment #7. I believe EDX mapping results of the samples may also provide more valuable information about the modification quality.

► Authors agreed with the reviewer's opinion. However, at this time, the EDS analysis by using TEM was only possible with the point technique, there is a limit to the overall composition analysis. The object of TEM and EDS analyses was limited on the confirmation of Au NPs in mesoporous ZnO matrix, and darker Au NP image and EDS result including Au NPs can be sufficient for this purpose. Authors request a generous understanding of reviewer.

Comment #8. Please provide the Scherrer equation and size calculation process. Also only one Au NP was indicated in Fig. 1, which cannot provide a solid support for the Scherrer calculation results.

► According to the comments of the reviewer, the authors added Scherrer calculation as follows.

After revision (Results and discussion):

Using the Scherrer equation [23], the size of the Au NPs was calculated with the full-width at half-maximum value of the Au (111) peak. The Scherrer equation is shown below.

$$D = \frac{K\lambda}{\beta \cos \theta}$$

Here, β is the width of the observed diffraction line at its half intensity maximum, K is the shape factor set to 0.9, and λ is the wavelength of Cu $K\alpha$. The size of Au NPs in AZO and ZnO was

calculated to be 20.7 nm and 21.2 nm, respectively which matched well with the TEM observations in Fig. 1.

Comment #9. Error bars should be provided in Fig. 4-6 to indicate the statistical significance, otherwise it is hard to determine the observed differences.

► According to the comments of the reviewer, authors added the error bar in Figs. 4~6.

After revision (Fig. 4, Fig. 5, Fig. 6):

Fig. 4. The porosity values of the pristine mesoporous ZnO, Au NPs-ZnO, and Au NPs-AZO thin films.

Fig. 5. The Seebeck coefficient values of the pristine mesoporous ZnO, Au NPs-ZnO, and Au NPs-AZO thin films according to temperature.

Fig. 6. The electrical conductivity of the pristine mesoporous ZnO, Au NPs-ZnO, and Au NPs-AZO thin films according to temperature.

Reviewer: 2

Comments to the Author(s)

Comment #1. Figure 1 depicts the TEM image and EDS results of the Au NPs-AZO thin film, which reveals formation of Au nanoparticles only. How sample was prepared for TEM analysis? By ion milling or scratching-ff the material from Si/SiO₂ substrate, dispersing it in some solvent and drop casting it on Cu grid? TEM images of pristine mesoporous ZnO and Au-NPs-ZnO should be provided for better understanding. In addition, the EDAX spectrum shows presence of Zn, Au, O, and Cu atomic species. There is no peak due to Al. Why? It is better to provide the atomic percentage of these elements in tabular form.

► According to the comments of the reviewer, authors added sampling method in Experimental Procedure part as “TEM sampling was achieved by carbon coated Cu grid by dispersing the powder in ethanol instead of ion milling or scratching. The powder was formed by heat treatment of sol solution.”

Concerning with the TEM and EDS data of mesoporous ZnO for comparison with ZnO containing

Au NPs (Au NPs-AZO), the additional TEM and EDS analyses of mesoporous ZnO were carried out and the results were given as Fig. 1(a). Please refer to the responses of the Comments #6 and #7 of Reviewer 1.

The composition of Al, Au and Zn in starting precursors were given in Experimental procedure part as “the molar ratios of the Al and Au precursors to Zn precursor were fixed at 0.02 (i.e. the atomic ratio of both Al to Zn and Au to Zn was 2 at%)”. This is not the final composition ratio in the mesoporous Au NPs-AZO thin film but all the precursors were preserved in the film formation process without a loss and then the starting composition can be used for the overall composition. Concerning with the emission peak of Al, the emission peak from Al (around 1.5 KeV) should have been observed in the EDS data of Fig. 1(b). However as shown in the TEM and EDX data, Au NPs was focused during the observation and the emission intensity of Au would have been greatly exaggerated. Then as shown in Fig. 1(b), Au emission peak near 1.5 KeV probably cover the Al emission peak. In order to observe the emission peak of Al, authors should have done an extra EDS analysis on AZO. However, due to the time limit (revision in 3 weeks), author could not obtain EDS result for the AZO. Author request a generous understanding of reviewer. However, authors believe that the effect of Al-doping in mesoporous ZnO thin film such as changes in crystallinity of ZnO, pore structure of mesoporous ZnO, Seebeck coefficient, and electrical conductivity clearly demonstrated the presence of Al-dopant in mesoporous ZnO thin film.

After revision (Experimental Procedure):

TEM sampling was achieved by carbon coated Cu grid by dispersing the powder in ethanol instead of ion milling or scratching. The powder was formed by heat treatment of sol solution.

Comment #2. The authors have recently published similar work in *Materials Chemistry and Physics* 212 (2018) 499 (Ref 17). What is the novelty in the present work? Whether the earlier results (Ref 17) are superior (or inferior) to the present ones?

► The authors' previous study (Materials Chemistry and Physics, 212 (2018) 499 (Ref 18)) was about Au NPs incorporation in mesoporous ZnO thin film to enhance the thermoelectric property by an increase of electrical conductivity from the high electrical conductivity and surface plasmon effect of Au NPs. However, the incorporation of Au NPs had a deleterious effect because of pore structural collapse. By the way, an incorporation of Al-dopant in ZnO matrix induces a stress in thin film, which distorts the crystalline structure of ZnO and as a result, grain growth of ZnO is inhibited and the pore structure can be maintained. The grain growth of ZnO hexagonal wurtzite structure is inhibited because of the difference in the radius of Al^{3+} (0.054 nm) and Zn^{2+} (0.074 nm). That is to say, the pore structural collapse due to the incorporation of Au NPs can be depressed by doping with Al, while still maintaining the increase in electrical conductivity. When the ZnO structure is doped with Al, the electrical conductivity of Al-doped ZnO (AZO) was increased because of an increase in carrier concentration.

In this study, this complementary effect of Al doping and incorporating Au NPs in mesoporous ZnO thin film was investigated for the first time, for example, crystallization, pore structure, porosity, and thermoelectric property of mesoporous ZnO thin film.

The Seebeck coefficient and electrical conductivity of Au NPs ZnO thin films in previous and present experiments showed similar values. But the complementary effect of Al doping and incorporating Au NPs in mesoporous ZnO thin film was found with XRD, GISXAS, Seebeck coefficient, and electrical conductivity measurements data in this study as superior thermoelectric properties of Au NPs-AZO than those of Au NPs-ZnO.

Comment #3. What is effect of size of Au NPs on ZT? As the surface plasmon resonance depends on size and shape of the NPs, will it influence the ZT?

► In nanocomposite, the size of the nanoparticle is a factor that could significantly change the ZT.

Generally, as the size of the nanoparticle decreases, the grain boundary increases, and this boundary could scatter the electron and phonon, affecting the electrical conductivity / thermal conductivity. In this study, free electrons are generated by surface plasmon resonance to improve the electrical conductivity.

However, as reviewer commented, as the surface plasmon resonance depends on size and shape of the NPs, it influences the ZT. That is because the electrical conductivity and thermal conductivity decrease as the size of the nanoparticle decreases [Ref#1]. In the previous paper, electrical conductivity changes with the addition of Au nanoparticles was analyzed [17].

[Ref#1] M. R. Dirmyer, J. Martin, G. S. Nolas, A. Sen, and J. V. Badding, *Small* 5 (2009) 933-937.

[17] M.-H. Hong, C.-S. Park, W. Han, B. Yoo, and H.-H. Park, *Chemistry Letters* 44 (2015) 1485-1487.

Comment #4. The manuscript contains a lot many typos and grammatical mistakes. The manuscript should be re-written carefully, avoiding vague statements. (For example, the statement" the pore structural distortion due to the Au NP incorporation in mesoporous ZnO to enhance electrical conductivity was minimized by Al doping" seems unclear.)

► The sentence was corrected and the entire manuscript was English corrected by English edition service. Authors hope no more typos and grammatical mistakes in the revised manuscript.